# Probiotics for Preventing Necrotizing Enterocolitis in Preterm Infants: A Network Meta-Analysis

**DOI:** 10.3390/nu13010192

**Published:** 2021-01-09

**Authors:** Isadora Beghetti, Davide Panizza, Jacopo Lenzi, Davide Gori, Silvia Martini, Luigi Corvaglia, Arianna Aceti

**Affiliations:** 1Neonatal Intensive Care Unit, AOU Bologna, Department of Medical and Surgical Sciences (DIMEC), University of Bologna, 40138 Bologna, Italy; isadora.beghetti@studio.unibo.it (I.B.); silvia.martini9@unibo.it (S.M.); luigi.corvaglia@unibo.it (L.C.); arianna.aceti2@unibo.it (A.A.); 2Department of Biomedical and Neuromotor Sciences (DIBINEM), University of Bologna, 40126 Bologna, Italy; jacopo.lenzi2@unibo.it (J.L.); davide.gori4@unibo.it (D.G.)

**Keywords:** necrotizing enterocolitis, preterm infants, probiotics, systematic review, network meta-analysis

## Abstract

Background: Recent evidence supports a role of probiotics in preventing necrotizing enterocolitis (NEC) in preterm infants. Methods: A systematic review and network meta-analysis of randomized controlled trials (RCTs) on the role of probiotics in preventing NEC in preterm infants, focusing on the differential effect of type of feeding, was performed following the Preferred Reporting Items for Systematic Reviews and Meta-Analyses (PRISMA) guidelines. A random-effects model was used; a subgroup analysis on exclusively human milk (HM)-fed infants vs. infants receiving formula (alone or with HM) was performed. Results: Fifty-one trials were included (10,664 infants, 29 probiotic interventions); 31 studies (19 different probiotic regimens) were suitable for subgroup analysis according to feeding. In the overall analysis, *Lactobacillus acidophilus* LB revealed the most promising effect for reducing NEC risk (odds ratio (OR), 0.03; 95% credible intervals (CrIs), 0.00–0.21). The subgroup analysis showed that *Bifidobacterium lactis* Bb-12/B94 was associated with a reduced risk of NEC stage ≥2 in both feeding type populations, with a discrepancy in the relative effect size in favour of exclusively HM-fed infants (OR 0.04; 95% CrIs <0.01–0.49 vs. OR 0.32; 95% CrIs 0.10–0.36). Conclusions: *B. lactis* Bb-12/B94 could reduce NEC risk with a different size effect according to feeding type. Of note, most probiotic strains are evaluated in few trials and relatively small populations, and outcome data according to feeding type are not available for all RCTs. Further trials are needed to confirm the present findings.

## 1. Introduction

Necrotizing enterocolitis (NEC) is a significant cause of morbidity and mortality in preterm infants with low birth weight (BW) and gestational age (GA). Recent meta-analyses estimated that 7% of low BW infants in neonatal intensive care units are likely to develop NEC [1]; mortality varies from 10% to 30% and has remained largely unchanged since the initial disease description several decades ago.

Evidence of the protective role of human milk (HM) against NEC in preterm infants is robust [2,3,4,5]. In recent years, the role of probiotic administration in preventing NEC has been also investigated through observational studies and randomized controlled trials (RCTs), whose results have been summarized in numerous systematic reviews and meta-analyses [6,7,8,9,10,11,12,13,14]. However, these studies have failed to provide clinically meaningful recommendations [15].

To overcome traditional meta-analytical approach limitations, van den Akker et al. recently performed a strain-specific network meta-analysis (NMA) on probiotics’ effects in reducing mortality and morbidity in preterm infants [16]. Following updated evidence provided by this study, in 2020, the European Society of Paediatric Gastroenterology Hepatology and Nutrition (ESPGHAN) published a position paper [17] aimed at providing clinicians with recommendations on probiotic strains with proven efficacy and safety in preterm infants. In addition, a recent NMA [18] focused on the differential performance of multi- vs. single-strain probiotics in reducing mortality and morbidity in preterm infants [16].

Beyond strain specificity, current scientific literature about probiotic use in preterm infants lacks detailed information about potential significant moderators, including type of feeding [15]. It has been suggested that probiotics might be more effective in infants fed HM rather than formula milk (FM), with nutrition and supplemental probiotics acting together in preterm infants’ guts, leading to a differential effect of exogenous bacteria depending on the type of feeding [12,19,20].

Hence, we performed an updated systematic review and NMA investigating probiotics’ role in preventing NEC in preterm infants and focused on the potential moderation effect of feeding type.

## 2. Materials and Methods

### 2.1. Study Design, Search Strategy and Selection Criteria

The study protocol was designed by the authors AA, IB, DG and JL. A systematic review of published studies reporting the use of probiotics for NEC prevention in preterm infants, according to type of feeding, and a Bayesian NMA were performed in accordance with the PRISMA Extension Statement for Reporting of Systematic Reviews incorporating NMA—Appendix A [21].

Criteria for inclusion in the systematic review and NMA were the following: RCTs or quasi-RCTs involving preterm infants (gestational age <37 weeks) and reporting on NEC (any stage, according to the modified Bell’s staging criteria); enteral administration of any probiotic starting within one month of age; probiotic nomenclature available at least at a species level. Studies comparing probiotic treatment against placebo, usual care or head-to-head with a different probiotic regimen, including single- and multiple-strain probiotics, were considered eligible.

PubMed (http://www.ncbi.nlm.nih.gov/pubmed/) and the Cochrane Library (http://www.cochranelibrary.com/) were interrogated for studies published before 7th January 2020 (Appendix A). The review was restricted to studies involving human subjects; no language restriction was applied.

AA, IB and DP independently performed the literature search. Studies potentially eligible according to the inclusion criteria were identified from the abstracts; full texts of relevant studies were assessed for inclusion and their reference lists were searched for additional studies. Reference lists of previously published meta-analyses were screened to identify further eligible studies. Inconsistency was resolved through consensus.

### 2.2. Study Selection and Data Extraction

Study details (population, characteristics of probiotic and placebo, type of feeding and outcome assessment) were evaluated independently by IB, DP and AA. Data were extracted using standardized forms. Missing or incomplete data, including separate data for NEC incidence in infants receiving probiotics vs. controls according to type of feeding (exclusive HM vs. exclusive formula or mixed feeding), were requested by contacting the corresponding authors. If the corresponding author was unable to provide these data or did not reply to the email, the paper was excluded from the meta-analysis.

According to the International Scientific Association of Probiotics and Prebiotics and the statement by the Food and Agriculture Organization of the United Nations and the World Health Organization for probiotics [22], probiotic products were identified at the highest taxonomic level whenever possible and, to the best of our knowledge, we adhered to the latest nomenclature and taxonomic description. In this respect, *Lactobacillus sporogenes* was designated as *Bacillus coagulans*, *Bifidobacterium infantis* 35,624 as *Bifidobacterium longum* 35,624 and *Bifidobacterium bifidum* Bb-12 as *Bifidobacterium lactis* Bb-12. Since *Bifidobacterium lactis* Bb-12 and *Bifidobacterium lactis* B94 share many characteristics, these strains were analyzed together. When strain level was not available in the original reports, whenever possible, it was obtained through Internet searches (i.e., on probiotic manufacturers’ websites) on available product names and technical sheets or previously published meta-analyses.

### 2.3. Quality Assessment and Risk of Bias

Quality assessment and risk of bias critical appraisals were conducted using the Cochrane risk of bias tool for RCTs [23]. IB and DP assessed each study separately, with conflicts resolved through consensus. Funnel plots were used to investigate publication bias. In addition, assessment of body of evidence quality using the GRADE (Grading of Recommendations Assessment, Development and Evaluation) working group approach was performed by DG [24].

All authors had access to the study data and reviewed and approved the final manuscript.

### 2.4. Statistical Analysis

We conducted a random-effects Bayesian NMA to synthesize all available evidence on probiotics for NEC prevention in preterm infants and to obtain a comprehensive ranking of all available probiotic treatments. Given that most treatments were evaluated in a limited number of RCTs, to reduce the sparsity of the network and potential model instability, we chose to pool single treatments into categories according to single treatments’ characteristics. At first, multi-genus probiotics were included in the same treatment category; we then performed a secondary NMA on main grouped treatments to rank their efficacy. Individual probiotic interventions and corresponding treatment categories are summarized in Appendix A.

Network structure was explored using network diagrams. Summary effect sizes were expressed as posterior medians of odds ratios (ORs) with 95% credible intervals (CrIs). All pairwise comparisons were summarized in a “league table”, a square matrix containing all information about relative efficacy (ORs) and their uncertainty (95% CrIs) for all possible pairs of interventions. Forest plots were generated to inspect the relative effects of each treatment compared to placebos. Surface under the cumulative ranking curve (SUCRA) was calculated to rank the efficacy of the various treatments [25]. Furthermore, NMA was also performed according to the frequentist approach and P-scores were calculated to compare treatments. Inconsistency was assessed using unrelated mean-effects models.

We evaluated whether infant feeding could act as an effect modifier across treatment comparisons by carrying out a subgroup analysis on exclusively HM-fed infants vs. infants receiving formula (alone or in addition to HM), and by informally comparing the magnitudes of effects and rankings.

We fit the NMA model using WinBUGS 1.4.3 [26] and visualized results using Stata 15 software (StataCorp. 2017. Stata Statistical Software: Release 15. College Station, TX: StataCorp LLC) and the Microsoft Excel-based tool NetMetaXL. All analyses were duplicated in the frequentist framework using the netmeta package in R [27]. Classic pairwise forest plots were generated using RevMan (version 5.4). A detailed description of the statistical methods is provided in Appendix B.

## 3. Results

### 3.1. Search Results and Study Characteristics

The number of potentially relevant papers identified through the literature search was 715 (556 in MEDLINE via PubMed and 159 in Cochrane Library); 19 additional trials were retrieved by hand searching. After excluding duplicates, 648 records were screened, of which 576 were excluded due to not meeting the inclusion criteria. Finally, 51 studies assessing the effects of 29 different probiotic interventions were eligible for inclusion in the systematic review and primary NMA [28,29,30,31,32,33,34,35,36,37,38,39,40,41,42,43,44,45,46,47,48,49,50,51,52,53,54,55,56,57,58,59,60,61,62,63,64,65,66,67,68,69,70,71,72,73,74,75,76,77,78] (Figure 1).

Only 18 studies reported NEC according to feeding type during the study period: ten studies reported NEC in exclusively HM-fed infants [29,33,40,51,53,62,64,70,74,76], either the mother’s own (MOM) or donor human milk (DHM), five studies included exclusively formula-fed infants [34,36,55,69,77], one study included infants who received HM plus FM [46] and two studies reported outcome according to different feeding types [37,57]. Separate data according to type of feeding were provided by e-mail for 13 of these studies [28,39,44,45,48,56,58,63,66,67,68,72,75]. Thus, 31 studies assessing the effects of 19 different probiotic interventions were suitable for inclusion in the subgroup NMA focusing on feeding.

Details of the included studies and NEC rates in the probiotic and in the placebo/control groups are reported in Appendix A. Identified probiotic strains and corresponding treatment groups are summarized in Appendix A. Not all probiotic strains were retrieved.

### 3.2. Quality Assessment and Risk of Bias

The quality assessment of the studies included in the NMA is shown in Appendix A. Few included studies were at low risk of bias on all components. Appendix A also shows the body of evidence assessment using the GRADE working group approach. A visual inspection of the evidence base by means of funnel plot (Appendix A) did not show any clear publication bias.

### 3.3. Probiotics and NEC: Overall Population

Figure 2 shows the network graph comparing probiotic categories for the prevention of NEC in the overall population, regardless of type of feeding. The network geometry is based on 13 treatment categories, assessed in 42 studies, versus the common comparator placebo, including 9658 patients. There was only one trial performing a direct head-to-head comparison between two probiotic treatments [44]. Nine studies, including 1006 infants, were initially excluded as there were zero events in both the placebo and intervention groups (Appendix A) [43,50,57,60,71,72,75,77,78].

According to SUCRA values of each treatment group (Figure 3A), *Lactobacillus acidophilus* LB ranked as the best supplementation option for preventing NEC in preterm infants; other high-ranking options were *B. lactis* Bb-12 OR B94, *B. longum* BB536, *L. reuteri* DSM 17938/ATCC 55,730 and the multi-genus probiotic group. These findings are consistent with the rankograms illustrated in Appendix A.

However, as shown in Figure 3B, the odds ratio of *B. longum* BB536 showed a wide CrI including one (zero on the log scale). On the other hand, the ORs and CrIs of *Lactobacillus acidophilus* LB, *B. lactis* Bb-12 OR B94, *L. reuteri* DSM 17938/ATCC 55,730 and the multi-genus group confirmed that these treatments were associated with a reduced risk of all stages of NEC. A league table with all pairwise comparisons between the competing treatments (ORs with 95% CrIs) is shown in Appendix A. No evidence of inconsistency in the network was documented (Appendix A). Results of the subgroup NMA conducted to rank single treatments included in the multi-genus probiotic group are shown in Appendix A.

The main findings of the Bayesian analysis agreed with results from the frequentist approach, where a similar treatment ranking according to the P-scores was found (Appendix A). The results proved to be robust also when trials with no events in both arms were included in the analysis (Appendix A).

### 3.4. Probiotic and NEC According to Feeding Type

An NMA focused on feeding type was also performed, including only studies for which outcome data according to feeding type were available. The overall network geometry was based on 13 treatment categories, assessed in 27 studies, versus the common comparator placebo, including 4173 patients. Four double-zero trials, including 378 infants, were initially excluded from the primary Bayesian meta-analysis [57,72,75,77], as well as an additional treatment investigated by Oshiro and colleagues (B.bBBG, *B. breve* BBG-001) [57].

*Lactobacillus acidophilus* LB showed the highest SUCRA value, followed by *B. longum* BB536, *B. breve* M-16V, *L. reuteri* 17938, *B. lactis* Bb-12 OR B94 and the multi-genus group (Figure 4A). However, as shown in Figure 3B, the ORs of *Lactobacillus acidophilus* LB, *B. longum* BB536 and *B. breve* M-16V had wide CrIs that included one (zero on the log scale). On the other hand, the distributions of the ORs of *L. reuteri* 17938, *B. lactis* Bb-12 OR B94 and the multi-genus group indicated that these treatments were associated with a reduced risk of NEC (Figure 4B). The results proved to be robust also when evidence from trials with no events in both arms was incorporated (Appendix A).

The results of the subgroup NMA conducted to rank single treatments included in the multi-genus probiotic group are shown in Appendix A.

The only interventions that were tested both in exclusively HM-fed infants and non-exclusively HM-fed infants were *B. lactis* Bb-12 OR B94, *L. reuteri* DSM17938, *L. rhamnosus* GG ATCC53103, *Saccharomyces boulardii* CNCM-I-745, *B. longum* 35,624 + *L. rhamnosus* GG and the multi-genus probiotic group. *B. lactis* Bb-12 OR B94, *L. acidophilus* and *L. reuteri* DSM17938 respectively showed the three best SUCRA values, followed by the multi-genus group (Figure 5A). However, according to the forest plot of relative effect sizes compared to placebo (Figure 5B), only *B. lactis* Bb-12 OR B94, based on one trial [39] with 91 infants, *L. reuteri* DSM17938, based on two trials [67,76] with 194 infants, and the multi-genus group, based on eight trials [29,33,44,51,62,63,64,68] with 1040 infants, were associated with a reduced risk of stage ≥ 2 NEC in exclusively HM-fed preterm infants.

A classic pairwise forest plot was constructed to explore specific probiotic treatments included in the multi-genus group (Appendix A). The OR distributions of *B. bifidum + B. infantis + B. longum + L. acidophilus* (based on one RCT [64] with 186 infants), *B. longum* R00175 + *L. helveticus* R0052 + *L. rhamnosus* R0011 + *Sa. boulardii* CNCM-I-1079 (based on two RCTs [29,68] with 254 infants) and *B. infantis* ATCC15697 + *L. acidophilus* ATCC4356 (based on one RCT [51] with 367 infants) indicated that these treatments were associated with a reduced risk of stage ≥ 2 NEC in exclusively HM-fed preterm infants.

Among the treatments evaluated in non-exclusively HM-fed preterm infants (Figure 5A–B) *B. longum* 35,624 + *L. rhamnosus* GG had the best SUCRA values; other efficacious options appeared to be *B. longum* BB536, *L. reuteri* DSM17938, the multi-genus probiotic group, *B. lactis* Bb-12 OR B94 and *L. rhamnosus* LGG ATTC53103. However, the ORs of *B. longum* 35,624 + *L. rhamnosus* GG, *L. rhamnosus* LGG ATTC 53103, the multi-genus probiotic group and *B. longum* BB536 showed a wide CrI that included one (zero on the log scale). On the other hand, the ORs of *B. lactis* Bb-12 OR B94, based on four trials [39,45,55,69] with 355 infants, and *L. reuteri* DSM17938, based on four RCTs [36,46,48,56] with 631 infants, indicate that these treatments were associated with a reduced risk of any-stage NEC in non-exclusively HM-fed preterm infants. Only *B. lactis* Bb-12 OR B94 was significantly associated with a reduced risk of stage ≥ 2 NEC, since one RCT evaluating *L. reuteri* DSM17938 also included stage 1 NEC [46].

Among treatments associated with much certainty with a reduced risk of NEC in both feeding sub-populations, we found a discrepancy in the relative effect size of *B. lactis* Bb-12 OR B94 in favor of exclusively HM-fed infants, although there was some overlap in the 95% CrIs (OR 0.04 with 95% CrI <0.01–0.49 vs. OR 0.32 with 95% CrI 0.10–0.96).

## 4. Discussion

To our knowledge, this is the first comprehensive and updated review of available literature regarding probiotic supplementation for prevention of NEC according to feeding type.

Fifty-one RCTs including data about over 10,000 infants were included in our systematic review. Outcome data according to feeding type were available for 31 RCTs including over 4550 infants; only eight out of 29 probiotic treatments were studied in both exclusively and non-exclusively HM-fed infants.

Some differences were observed between exclusively HM-fed vs. non-exclusively HM-fed infants. Based on available data and on both direct and indirect evidence, *B. lactis* Bb-12/B94 was associated with a reduced risk of stage ≥ 2 NEC both in exclusively HM- and non-exclusively HM-fed preterm infants, with a discrepancy in the relative effect size in favor of patients fed exclusively HM. *B. bifidum + B. infantis + B. longum + L. acidophilus*, *B. infantis* ATCC15697 + *L. acidophilus* ATCC4356 and *B. longum* R00175 + *L. helveticus* R0052 + *L. rhamnosus* R0011 + *Sa. boulardii* CNCM-I-107 were also efficacious in exclusively HM-fed infants. *L. reuteri* DSM17938 was associated with a reduced risk of stage ≥ 2 NEC in exclusively HM-fed preterm infants and with a reduced risk of NEC at any stage in non-exclusively HM-fed infants.

The results of our main analysis, performed regardless of infants’ feeding characteristics, were only partially overlapping with the strain-specific NMA performed by the ESPGHAN working group [16] since we included different and updated RCTs, some of which evaluated new probiotic treatments [33,36,40,43,44,46,48,50,57,73,76]. Furthermore, we did not include studies where probiotics were used together with other products [79,80] and we focused on RCTs for which outcome data according to feeding type were available.

Recently, an NMA focused on the differential performance of multi-strain vs. single-strain probiotics in reducing mortality and morbidity in preterm infants was published [18]. The results of our main analysis confirm the efficacy of some interventions highlighted by that paper, as the authors found that combinations of one or more *Lactobacillus* spp. and one or more *Bifidobacterium* spp., *Bifidobacterium lactis* and *Lactobacillus reuteri* significantly reduced severe NEC. However, some differences should be acknowledged. We chose to identify probiotics at a higher taxonomic level and to group probiotic treatments differently; moreover, included RCTs reporting on NEC are not completely overlapping.

Beyond probiotics’ efficacy, safety issues and possible probiotic side effects should also be considered when choosing a probiotic product to be administered to preterm infants. In this respect, the ESPGHAN panel on probiotics and preterm infants has recently issued a conditional recommendation not to provide probiotic strains that produce D-lactate (i.e., *L. reuteri* DSM17938 or *L. acidophilus* NCO1748), as their potential risks or safety have not been adequately studied in preterm infants and remain uncertain.

Some limitations of the present paper need to be acknowledged: both for the overall population and for the subgroup population for which feeding data were available, the number of probiotic treatments was considerable, most of the treatments were only evaluated in one or two trials and only few strains were tested in at least four RCTs. As a result, most probiotic interventions were evaluated in small experimental populations. To cope with network sparseness and potential instability, we chose to group probiotic interventions into treatment categories according to probiotics’ characteristics. Furthermore, the available information on the highest-risk infants, such as those with extremely low birth weight (ELBW) or intrauterine growth retardation, was limited: actually, only five [28,50,54,57,70] of the 51 included studies included ELBW infants, and only three [28,57,70] studies reported outcome in ELBW infants according to feeding type.

As for study quality, the evidence base was frequently dependent on many studies with an unclear risk of bias for the various domains, with 12 studies having a high risk of bias in at least one domain. Of note, most of the studies with a high risk of bias did not contribute to the significant probiotic interventions. Most of the original RCTs did not report NEC as a primary outcome and appeared to be underpowered. Therefore, the generalizability and clinical applicability of our results require further confirmation.

## 5. Conclusions

We aimed to provide an overview of all published evidence on the use of probiotics for prevention of NEC in preterm infants according to infant feeding characteristics.

Some interesting preliminary results were produced. Nonetheless, these results must be interpreted in light of some limitations, as most probiotic strains were studied few times and in small experimental populations. Therefore, the available data did not allow to provide strong recommendations on the most promising probiotic intervention(s) and to clarify the relationship between NEC prevention and type of feeding in preterm infants receiving probiotics. There is a need for adequately powered studies on probiotic supplementation in preterm infants describing in greater detail the feeding characteristics of included infants, with the aim of limiting potential confounding factors or assessing the role of potential effect modifiers (i.e., percentage of MOM, duration of exclusive MOM, characteristics of HM fortification, etc.). To overcome the impracticability of randomizing preterm infants according to type of feeding, it appears reasonable to restrict inclusion criteria to infants fed homogeneously (MOM, DHM or formula). Moreover, stringent controls on the production of probiotics and attention to correct product identification at the highest taxonomic level are needed not only for research purposes but also for clinical implementation.

## Figures and Tables

**Figure 1 nutrients-13-00192-f001:**
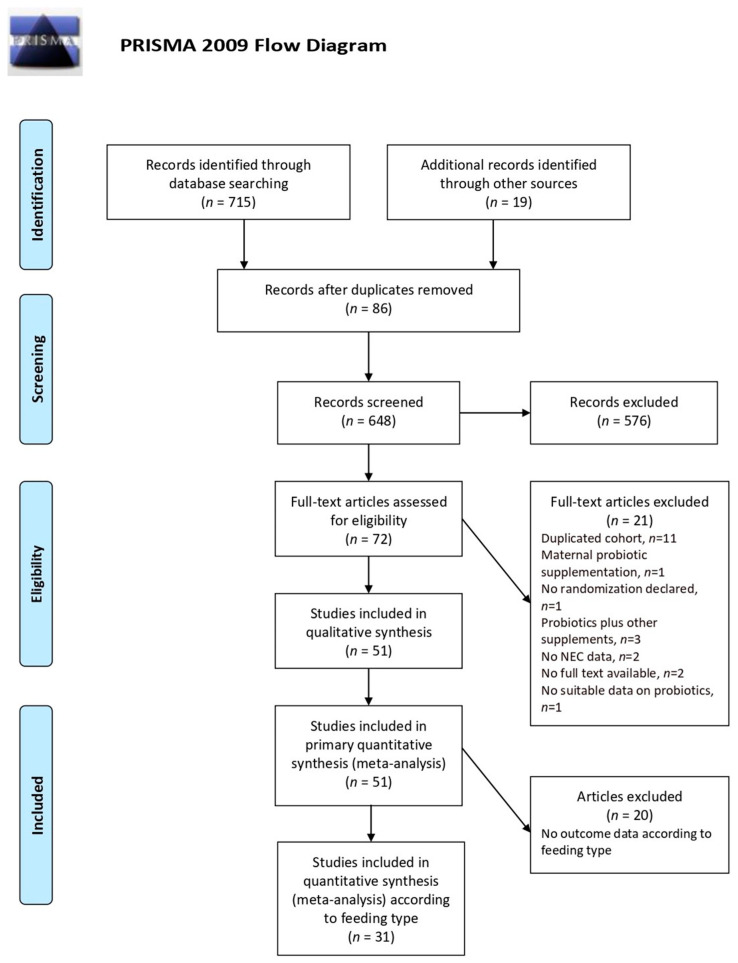
Flow diagram of search strategy and study selection.

**Figure 2 nutrients-13-00192-f002:**
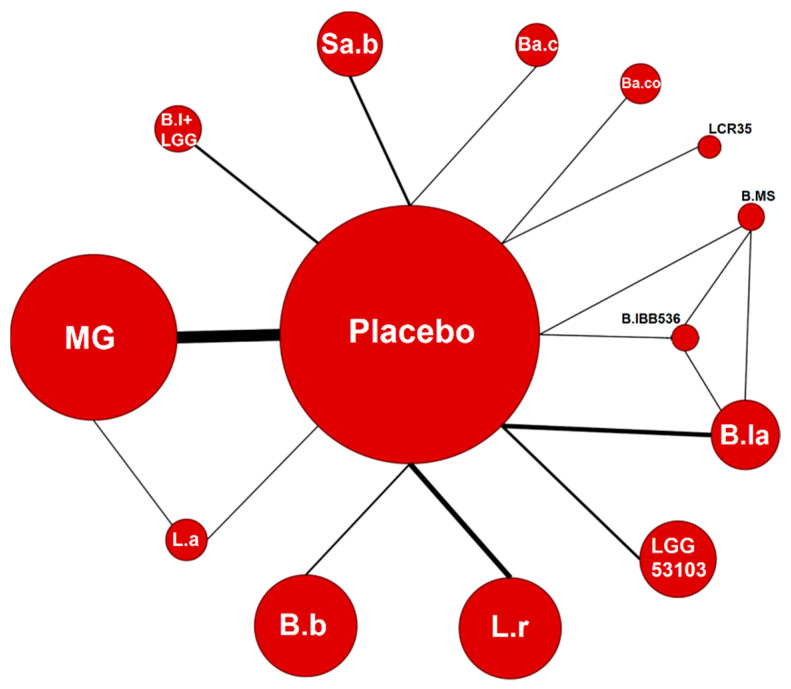
Network diagram. The area of the nodes is based on the total number of patients for each treatment among all trials. The thickness of the lines represents the total number of studies comparing the two treatments/nodes. B.b: *B. breve* BBG YIT4010, *B. breve* BBG-001, *B. breve* M-16V; B.la: *Bifidobacterium lactis* Bb-12 OR B94; B.lBB536: *B. longum* BB536; L.r: *Lactobacillus reuteri* DSM 17938, *L. reuteri* ATCC 55730; LGG53103: *L. rhamnosus* GG ATCC 53103; L.a: *L. acidophilus* LB; LCR35: *L. casei* var. *rhamnosus* (LCR 35); Ba.co: *Bacillus coagulans* (*L. sporogenes*); Ba.c: *Ba. clausii* (four strains); Sa.b: *Saccharomyces boulardii* CNCM I-745, *Sa. boulardii* CNCMI-3799; B.MS: *B. lactis* Bb-12 + *B. longum* BB536; B.l + LGG: *B. longum* 35,624 + *L. rhamnosus* GG, *B. longum* BB536 + *L. rhamnosus* GG; MG: multi-genus probiotic group.

**Figure 3 nutrients-13-00192-f003:**
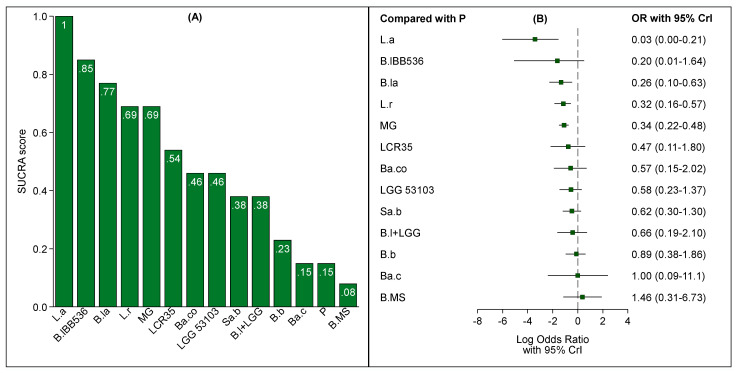
Bar chart of surface under the cumulative ranking curve (SUCRA) scores (**A**) and forest plot of relative effect sizes compared to placebo (**B**) for each treatment under study.

**Figure 4 nutrients-13-00192-f004:**
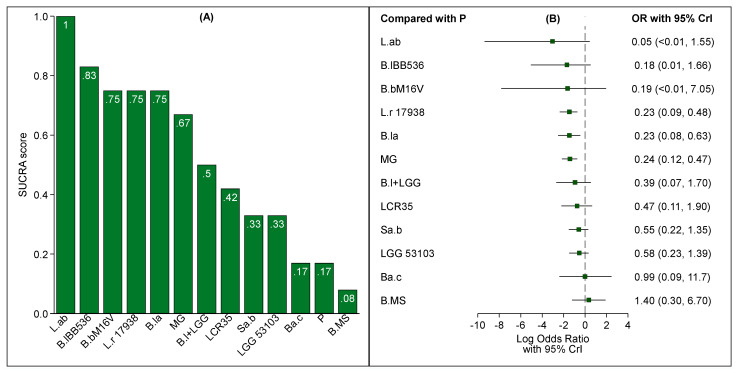
Bar chart of SUCRA scores (**A**) and forest plot of relative effect sizes compared to placebo (**B**) for each treatment under study. Only trials that made information on type of feeding available were investigated.

**Figure 5 nutrients-13-00192-f005:**
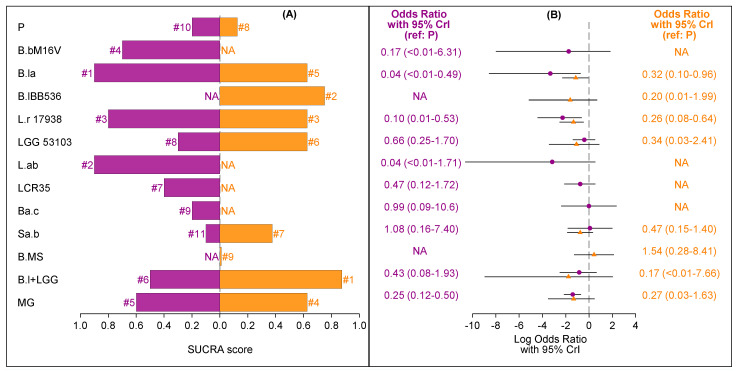
Bar chart of SUCRA scores (**A**) and forest plot of relative effect sizes compared to placebo (**B**) for each treatment under study by type of feeding (purple: exclusively human milk (HM); orange: exclusively formula milk and non-exclusive HM). NA, not available.

## Data Availability

Data sharing not applicable.

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
