# Peer review of "Probiotics for Preventing Necrotizing Enterocolitis in Preterm Infants: A Network Meta-Analysis"

_nutrients, 2021, doi:10.3390/nu13010192_

Round 1
Reviewer 1 Report
There are major and minor points, typos or curiosities, that should be revised, explained and, if necessary, corrected.
Major points:
The search strategy:
If a systematic review is considered, why don't more databases such as Embase, Scopus, Web of Science, etc., be queried?
When querying the MEDLINE database, (via Pubmed) Why are truncated descriptors used (higher noise) when Mesh Necrotizing Enterocolitis exists (D020345)? It´s also occurs with probiotic (D019936). Would it not have been more appropriate to use also, in the search strategy, the descriptor in [title/abstract]? For example, necrotizing enterocolitis [title/abstract].
If the review is restricted to human studies, why is NOT HUMANS written in the search strategy? Would the results and conclusions be reliable?
I think the search strategy should be reviewed and some Boolean operators should be placed correctly “…..OR enterocolitis) AND NEC) AND probiotics….”
Minor points:
Line 97 : “…internet searches” Which websites were investigated?
Line 136: “….was 715 (556 in PubMed)…”. Could it be more correct 556 in MEDLINE (via PubMed)?
Line 139: [28, 29,…..30,…..31….32,…33,.34-37] Is it correctly sequenced?. If the references are consecutive and more than two, they are also described in the same bracket only the first and last and separated by a hyphen.
If the selection criterion was the review of clinical studies, why was CONSORT 2010 statement not used to quality assessment?
In the Figure 1, what was the reason for excluding 576 articles?
Line 289: You should check the order of the bibliographic references
The study included in the review, Awad, 2010. The probiotic strain used is Lactobacillus acidophilus LB. Shouldn't it be excluded from the review because it uses killed forms? Probiotics are living microorganisms.
Author Response
Reviewer 1
There are major and minor points, typos or curiosities, that should be revised, explained and, if necessary, corrected.
Major points:
The search strategy:
1. If a systematic review is considered, why don't more databases such as Embase, Scopus, Web of Science, etc., be queried?
We are grateful for this comment which allows us to better explain the reasons for our choices and exclusion. The reason for Embase exclusion was to be as much time efficient as possible. Previous papers (Rollin L, Darmoni S, Caillard JF, Gehanno JF. Searching for high-quality articles about intervention studies in occupational health--what is really missed when using only the Medline database? Scand J Work Environ Health. 2010 Nov;36(6):484-7. doi: 10.5271/sjweh.3082. Epub 2010 Aug 25. PMID: 20740265.) have demonstrated that the vast majority (90%) of RCTs in Cochrane Reviews can be found in PubMed. In addition, another paper gives the lead for RCT retrieval to Cochrane Register for Trials and PubMed (https://pubmed.ncbi.nlm.nih.gov/15095765/). These were the main reasons which led us to consider only CCRT and PubMed as primary sources for trial retrieval for an exhaustive but time sparing search. The other databases such as Scopus and Web of Science were excluded for the same reasons and due to their relative low specificity for additional RCTs retrieval.
- When querying the MEDLINE database, (via Pubmed) Why are truncated descriptors used (higher noise) when Mesh Necrotizing Enterocolitis exists (D020345)? It´s also occurs with probiotic (D019936). Would it not have been more appropriate to use also, in the search strategy, the descriptor in [title/abstract]? For example, necrotizing enterocolitis [title/abstract].
Thanks for this comment. As the Reviewer knows, every MeSH term is evoked by entry terms. For example, as far as Necrotizing Enterocolitis (MeSH term – D020345 https://www.ncbi.nlm.nih.gov/mesh/?term=necrotizing+enterocolitis) is concerned, the entry term “necrotizing enterocolitis” automatically evokes the MeSH, as well as the term is searched in the title and the abstract. Due to our willingness to be the most exhaustive as possible, we decided to put terms in this way. The same consideration can be applied for probiotic (D019936).
If the review is restricted to human studies, why is NOT HUMANS written in the search strategy? Would the results and conclusions be reliable?
Thanks for this comment which allows to better explain our choices. We used, as shown in the complete string, a small string at the end: “NOT (animals [MH] NOT humans [MH])” to exclude those articles on animal diseases only. The use in PubMed of this sub-string means that all papers with the MeSH term “animals” and without the MeSH term “humans” will be excluded. The same result can be obtained by writing (NOT animals [MH] NOT (animals [MH] AND humans [MH])). This allowed to better focus our search on papers on humans, and we think that results and conclusion are even more reliable in this way rather than with the simple use of “NOT humans”.
I think the search strategy should be reviewed and some Boolean operators should be placed correctly “…..OR enterocolitis) AND NEC) AND probiotics….”
Thanks for this suggestion. Anyway, we claim to put the Boolean operators in this order as we wanted all the “probiotics” terms to be searched as a separate parenthesis. On the other side, NEC should be combined and associated with the “AND” to the other terms which might be too much sensitive.
Minor points:
5. Line 97 : “…internet searches” Which websites were investigated?
We thank the Reviewer for this question. In order to retrieve probiotics’ strains, we investigated probiotics’ manufacturer website when available. Additionally, we checked for nomenclature consulting the List of Prokaryotic Names with a Standing in Nomenclature (http://www.bacterio.net/).
Line 136: “….was 715 (556 in PubMed)…”. Could it be more correct 556 in MEDLINE (via PubMed)?
We thank the Reviewer for this comment. “ (556 in PubMed)” has been replaced by (556 in MEDLINE via PubMed)”.
Line 139: [28, 29,…..30,…..31….32,…33,.34-37] Is it correctly sequenced?. If the references are consecutive and more than two, they are also described in the same bracket only the first and last and separated by a hyphen.
We agree with the Reviewer that the line 141 reference should be described in the same bracket citing only the first and last reference separated by a hyphen. The mistake was generated by the software used for reference management and has not been solved.
If the selection criterion was the review of clinical studies, why was CONSORT 2010 statement not used to quality assessment?
Thanks for the comment. We are aware of the CONSORT 2010 statement, but, following the Cochrane methodology (https://handbook-5 1.cochrane.org/chapter_8/8_assessing_risk_of_bias_in_included_studies.htm), we preferred to use the Risk of Bias Tool (https://methods.cochrane.org/bias/resources/rob-2-revised-cochrane-risk-bias-tool-randomized-trials). In fact, we were more interested in finding the strength or pitfalls of the papers rather than the reporting errors, which might be bitterly used when a researcher is planning to write down an RCT rather than assessing it ex-post.
In the Figure 1, what was the reason for excluding 576 articles?
In the Figure 1, 576 articles were excluded as they did not meet the inclusion criteria reported in lines 67-70, 75. Specifically, they were reviews or meta-analyses, they were preclinical studies, not RCTs or did not involve preterm infants. This has been added in the text.
Line 289: You should check the order of the bibliographic references
We thank the Reviewer for his observation. Line 289 references sequence error has been solved.
The study included in the review, Awad, 2010. The probiotic strain used is Lactobacillus acidophilus LB. Shouldn't it be excluded from the review because it uses killed forms? Probiotics are living microorganisms.
We thank the Reviewer for this precious comment. According to recent evidence, the probiotic effect of beneficial microbes does not rely necessarily on their viability, but their interaction with the host might be based on the capacity of human cells to recognise specific bacterial components or products, giving rise to responses that commonly involve the mucosa-associated lymphoid tissue and, therefore, the immune system. The term “para-probiotics” or “ghost probiotics” has been proposed for “non-viable (more often heat-inactivated) microbial cells (intact or broken) or crude cell extracts (i.e., nucleic acids, cell-wall components), which, when administered (orally or topically) in adequate amounts, confer a benefit on the human or animal consumer” (Taverniti V, Guglielmetti S. The immunomodulatory properties of probiotic microorganisms beyond their viability (ghost probiotics: proposal of paraprobiotic concept). Genes Nutr. 2011 Aug;6(3):261-74. doi: 10.1007/s12263-011-0218-x). At present, few probiotic species, including strains of both Lactobacillus and Bifidobacterium, have been studied in their inactivated form. Growing interest for para-probiotics relies on the fact that these agents could provide their beneficial effects without the potential risk associated with the administration of live microorganisms, especially in vulnerable individuals such as preterm infants (Deshpande G, Athalye-Jape G, Patole S. Para-probiotics for Preterm Neonates-The Next Frontier. Nutrients. 2018 Jul 5;10(7):871. doi: 10.3390/nu10070871). For this reason, we chose to include in our review the study by Awad et al.
Reviewer 2 Report
This study reviewed and meta-analyzed the preventive effects of probiotics on necrotizing enterocolitis (NEC) in preterm infants focusing on the type of feeding.
The results provide significant information for assessing the beneficial effects of probiotics. However, the actual dataset of clinical trial using respective probiotics strain is relatively small, which should be acknowledged in conclusion and abstract session.
Minor points:
- The quality of Figure 2 should be improved. It is hard to capture the meaning of the network.
- L118: what's the meaning of a leaque table? Refer the type and number of the table data.
- To help the readers understand the results, all the results section should provide the summarized significance of represented strains from the authors' meta-analysis.
- The name of bacterial strains should not be an Italic characters (only Genus and Species name has to be presented using Italics). And there are typos should be corrected.
Author Response
Reviewer 2
This study reviewed and meta-analyzed the preventive effects of probiotics on necrotizing enterocolitis (NEC) in preterm infants focusing on the type of feeding.
The results provide significant information for assessing the beneficial effects of probiotics. However, the actual dataset of clinical trial using respective probiotics strain is relatively small, which should be acknowledged in conclusion and abstract session.
We thank the Reviewer for this comment. Both for the overall population and for the subgroup population for which feeding data were available, the number of probiotics treatments was considerable, and most of the treatments were only evaluated in one or two trials. Consequently, most probiotic strains were evaluated in small experimental populations. We acknowledged this limitation also in the conclusion and abstract section.
Minor points:
- The quality of Figure 2 should be improved. It is hard to capture the meaning of the network.
We agree with the Reviewer that the labels of Figure 2 are barely readable. We have enlarged font sizes, augmented colour contrast, and provided a new version of Figure 2.
- L118: what's the meaning of a league table? Refer the type and number of the table data.
Thank you for your suggestion. We have replaced the sentence «All pairwise comparisons were summarized in a league table, a matrix containing the estimated effect for each comparison» (lines 118–120) with the following one, which should be more explanatory: «All pairwise comparisons were summarized in a “league table”, a square matrix containing all information about relative efficacy (ORs) and their uncertainty (95% CrIs) for all possible pairs of interventions». Moreover, to clarify where this table is, we have changed the sentence «All pairwise comparisons between the competing treatments (ORs with 95% CrIs) are shown in Supplemental Figure 4» (lines 194–196) with «A league table with all pairwise comparisons between the competing treatments (ORs with 95% CrIs) is shown in Supplemental Figure 4».
- To help the readers understand the results, all the results section should provide the summarized significance of represented strains from the authors' meta-analysis.
Thank you for raising this point. As a summary presentation of results, we have presented the SUCRA scores, which rank the efficacy of all treatments included in the network meta-analysis. Another and probably more familiar tool would be the forest plot, where the significance of each strain can be inferred quite nimbly by looking the 95% credible interval—if it crosses the null value (0 or 1, depending on the scale), there is no systematic difference between treatments. Another useful presentation of results is the league table, that should be clearer and more visible now (see comment above). We think that the Methods and Results section are already rich and adding more summary measures might worsen the read.
- The name of bacterial strains should not be in Italic characters (only Genus and Species name has to be presented using Italics). And there are typos should be corrected.
We thank the Reviewer for this suggestion. Bacterial strains names have been corrected using non-Italics characters in the manuscript, in the supplemental material as well as in Table 1.
Reviewer 3 Report
Beghetti et al present a Network Meta-Analysis on probiotics for prevention of Necrotizing Enterocolitis in Preterm Infants. They conducted a concise review of the literature and found that B. lactis (Bb-12/B94) could reduce NEC risk.
The work gives a very nice summary of RCTs on this topic and adds important information.
Author Response
Reviewer 3
Beghetti et al present a Network Meta-Analysis on probiotics for prevention of Necrotizing Enterocolitis in Preterm Infants. They conducted a concise review of the literature and found that B. lactis (Bb-12/B94) could reduce NEC risk.
The work gives a very nice summary of RCTs on this topic and adds important information.
We would like to warmly thank Reviewer 3 for this very nice comment to our paper.